# Synovial Fluid Cytokines, Chemokines and MMP Levels in Osteoarthritis Patients with Knee Pain Display a Profile Similar to Many Rheumatoid Arthritis Patients

**DOI:** 10.3390/jcm10215027

**Published:** 2021-10-28

**Authors:** Richard T. Meehan, Elizabeth A. Regan, Eric D. Hoffman, Molly L. Wolf, Mary T. Gill, James L. Crooks, Prashant J. Parmar, Richard A. Scheuring, John C. Hill, Karin A. Pacheco, Vijaya Knight

**Affiliations:** 1Department of Medicines, Immunology Labs and Bioinformatics National Jewish Health, Denver, CO 80206, USA; regane@njhealth.org (E.A.R.); ehoffman@arthroventions.org (E.D.H.); wolfm@njhealth.org (M.L.W.); gillm@njhealth.org (M.T.G.); crooksj@njhealth.org (J.L.C.); pachecok@NJHealth.org (K.A.P.); 2Colorado School of Public Health, CU Anschutz School of Medicine, Aurora, CO 80045, USA; 3Department of Internal Medicine, National Jewish Health, Saint Joseph Hospital, Denver, CO 80218, USA; pparmar.2017@gmail.com; 4Flight Medicine, NASA-Johnson Space Center, Houston, TX 77058, USA; richard.a.scheuring@nasa.gov; 5CU Sports Medicine, Department of Orthopedic Surgery, University of Colorado, Denver, CO 80222, USA; john.hill@ucdenver.edu; 6Immunology Department, Children’s Hospital, Denver, CO 80045, USA; Vijaya.Knight@childrenscolorado.org

**Keywords:** synovial fluid, biomarkers, cytokines, osteoarthritis

## Abstract

Background: There are currently no effective disease-modifying drugs to prevent cartilage loss in osteoarthritis and synovial fluid is a potentially valuable source of biomarkers to understand the pathogenesis of different types of arthritis and identify drug responsiveness. The aim of this study was to compare the differences between SF cytokines and other proteins in patients with OA (*n* = 21) to those with RA (*n* = 27) and normal knees (*n* = 3). Methods: SF was obtained using ultrasound (US) guidance and an external pneumatic compression device. RA patients were categorized as active (*n* = 20) or controlled (*n* = 7) based upon SF white blood cell counts (> or <300 cells/mm^3^). Samples were cryopreserved and analyzed by multiplex fluorescent bead assays (Luminex). Between-group differences of 16 separate biomarker proteins were identified using ANOVA on log10-transformed concentrations with *p* values adjusted for multiple testing. Results: Only six biomarkers were significantly higher in SF from active RA compared to OA—TNF-α, IL-1-β IL-7, MMP-1, MMP-2, and MMP-3. Only MMP-8 levels in RA patients correlated with SF WBC counts (*p* < 0.0001). Among OA patients, simultaneous SF IL-4, IL-6, IL-8, and IL-15 levels were higher than serum levels, whereas MMP-8, MMP-9, and IL-18 levels were higher in serum (*p* < 0.05). Conclusion: These results support the growing evidence that OA patients have a pro-inflammatory/catabolic SF environment. SF biomarker analysis using multiplex testing and US guidance may distinguish OA phenotypes and identify treatment options based upon targeted inflammatory pathways similar to patients with RA.

## 1. Introduction

Osteoarthritis (OA) of the knee is very common, encompassing multiple etiologies with as yet no disease-modifying treatments prior to joint replacement. Knee OA is a growing problem worldwide due to aging populations, an increasing prevalence of obesity in developed countries, and even youth-related sports injuries [1,2,3]. It is also associated with increasing health care costs and disability—in the US, 32 million have symptomatic activity limiting OA, with estimated direct medical costs exceeding $100 billion [4,5].

Biomarkers that can accurately diagnose specific diseases, predict disease progression, monitor response to treatment and predict drug responsiveness in individual patients are urgently needed in the management of patients with arthritis [6]. A critical need is the accurate identification of inflammatory arthritis of the knee, permitting disease-modifying treatments that could prevent disease progression and the need for joint replacement.

Biomarkers in synovial fluid (SF) may hold the key to accurate diagnosis and treatment of knee arthritis. SF is secreted by synovial cells and reflects chondrocyte metabolism and matrix turnover [7,8]. The number of SF white blood cells (WBCs) is used routinely in clinical rheumatology to identify inflammatory arthritis although they remain non-specific indicators of increased inflammation within the joint space. In both rheumatoid arthritis (RA) and infectious arthritis, white blood cells can be rapidly destructive to cartilage due to the secretion of various metalloproteinases. OA represents, in part, dysfunction in essential cartilage matrix turnover, leading to a net loss of cartilage tissue [4,8]. Local inflammatory cytokines such as IL-1 and TNF-α are important in this process since a pro-inflammatory milieu can adversely affect chondrocyte health and the maintenance of healthy extracellular matrix (ECM) [9]. There is also growing evidence that the local production of pro-inflammatory cytokines IL-1-β, TNF-α, IL-6, IL-15, IL-17, and IL-18 is involved in the pathogenesis of OA [8]. Many of these cytokines can be synthesized by chondrocytes, osteoblasts, synovial tissues, and mononuclear cells. Thus, SF is a prime potential source for identifying key biomarkers of arthritis. Work in this area has been limited due to difficulty in collection of SF as many patients with progressive arthritis have sparse amounts of SF.

In contrast to the local milieu of SF, measuring specific cytokine, chemokine or MMP levels in the peripheral blood may not accurately reflect levels in the SF space since peripheral blood contains tens of thousands of separate proteins over a 12-fold dynamic range of concentration and many of these cytokines are produced by numerous extra-articular sites [7]. To date, measuring peripheral blood cytokines has not proven helpful at guiding clinicians to select the optimum disease-modifying anti-rheumatic drug (DMARD), biologic or Janus Kinase (JAK) inhibitor therapy in patients with inflammatory arthritis. Measuring the SF cytokine profile could identify which catabolic cytokines/chemokines and proteinases accelerate cartilage loss and therefore might be targets for therapeutic interventions. These may also more effectively distinguish which patients have a systemic immune-mediated inflammatory disease (IMID) vs. OA [9,10,11,12,13,14,15,16]. Biomarker-specific distinctions between these diagnoses may inform treatment decisions and would greatly advance personalized medicine strategies in the management of arthritis.

Our research objectives were to quantitate the levels of 16 selected SF cytokines and other key proteins in patients with OA compared to patients with active and controlled RA and normal knees. We also hoped to advance the field of SF biomarker research in OA by demonstrating that US guidance with external compression can facilitate obtaining adequate SF volumes for multiplex testing since many OA patients have very small SF volumes. We also wanted to identify if any SF cytokines correlated with WBC counts in the SF among patients with RA.

## 2. Materials and Methods

### 2.1. Subjects

All subjects consented and enrolled in an Institutional Review Board (IRB) approved study to have their SF cryopreserved (HS 2511, HS 3095, or HS 3179). SF was obtained from the knees of 51 individuals including patients with RA (*n* = 27), OA (*n* = 21) or healthy research subjects (*n* = 3) between 2015 and 2020. All patients were followed in our rheumatology practice at National Jewish Health (NJH), and underwent knee aspirations for diagnostic arthrocentesis or prior to an intra-articular (IA) glucocorticoid or hyaluronic acid product (HA) injection. For analysis, RA patients were divided into groups based upon their SF WBC counts. RA patients were designated as “RA active” if their SF WBC level was >300 WBCs/mm^3^ or “RA controlled” if their SF WBC count was <300 WBCs/mm^3^. In addition, a subset of our total OA patients (*n* = 11) were subjects in an investigator-initiated SF biomarker research study (HS 3179, ClinicalTrials.gov NCT 04093232) and had their knees aspirated prior to an intra-articular (IA) injection of HYADD 4 (Hymovis^TM)^) and provided simultaneous peripheral blood samples. The 3 normal subjects were recreational runners without knee pain and SF was aspirated as part of a study to investigate the role of knee unloading on cartilage health funded by CASIS/NASA (HS 3095). All patients and research subjects were evaluated or managed by a board certified rheumatologist to confirm the diagnosis of RA or OA.

To facilitate SF removal, knee aspirations were obtained in all but 3 subjects using an external pneumatic compression device (KneeTap^TM^, Arthroventions LLC Denver CO) inflated to 100 mmHg as previously described [17]. One patient had an excessive thigh girth and 2 RA patients had very large effusions. All SF samples were obtained using an 18 or 20 gauge needle and US guidance with a GE LOGIQ e ultrasound machine with images as noted in Figure 1. A serum RF result >14 units, measured on a Beckman Coulter AU 4480 analyzer, was considered positive and anti-CCP antibodies, measured using the QUANTA Lite CCP 3.1 ELISA assay, were considered positive if the result was >20 units. SF WBC quantification was performed by light microscopy using a hemocytometer at the time of sample preparation.

### 2.2. Sample Preparation and Analysis of SF and Serum Proteins

Samples were transferred within 60 min of collection to microcentrifuge tubes and centrifuged at 2000 rpm for 10 min. The supernatant was then aliquoted into 200 µL vials and stored at −80 °C until analyzed.

All analytes were measured by multiplex fluorescent bead (Luminex) assays using R & D Systems Inc. (Minneapolis, MN, USA) for TNF-α (Tumor Necrosis Factor alpha), MMP-8 (Matrix Metalloproteinase 8), MMP-2 (Matrix Metalloproteinase 2), MMP-1 (Matrix Metalloproteinase 1), IL-1 b (Interleukin 1 beta), IL-7 (Interleukin 7), IGFBP-3 (Insulin-Like Growth Factor-Binding Protein 3),MMP-3 (Matrix Metalloproteinase 3), IL-6 (Interleukin 6), IL-1 ra (Interleukin 1 receptor antagonist), IGFBP-4 (Insulin-Like Growth Factor-Binding Protein 4), IL-18 (Interleukin 18), IL-2 (Interleukin 2), IL-4 (Interleukin 4), G-CSF (Granulocyte Colony Stimulating Factor), and IL-15 (Interleukin 15). The bead multiplex assay was performed in 96-well plates. Samples were incubated with specific antibody-coated hard dyed fluorescent beads for 2 h, then washed twice and incubated with 100 mcL of biotinylated detection antibody for 1 h, washed twice and then incubated with 100 µL of streptavidin-RPE for 30 min. The plates were read on a Luminex 100 Bio-Plex suspension array system. Cytokine concentrations were calculated by reference to the standard curve for each analyte. The sensitivity of detection on the Bio-Plex is 3.2 pg/mL.

### 2.3. Statistical Analysis

For comparisons of cytokine levels between diagnoses groups, we used ANOVA with the cytokine concentration outcomes log10 transformed. *p*-values < 0.05 were considered significant and were adjusted for the number of analytes (*n* = 16) using the Bonferroni method. If values were missing for an individual subject for a specific analyte, then that subject was excluded from statistical analysis for that specific analyte. For the purpose of calculation, samples that exceeded the upper limit of the analytical measurement range or which were below the detection limit were assigned the upper limit value or lower limit value, respectively, for the respective cytokine, chemokine or MMP.

Linear regression analysis of WBC counts compared with log10-transformed analyte concentration was used to estimate associations between cytokine concentrations and WBC counts. WBC counts were also log10 transformed to decrease sensitivity to the few samples with the highest counts. All modeling was performed in the R language [18].

## 3. Results

Table 1 presents the demographic and treatment characteristics of the four subject groups. In the cohort of 20 active RA patients, only 58% were receiving immunomodulatory therapy with either prednisone, conventional disease-modifying anti-rheumatic disease drugs (cDMARDS), methotrexate, hydroxychloroquine, sulfasalazine or biologics. Some of these 20 RA patients were experiencing a flare of their systemic disease off medications, whereas others had refractory disease or were transitioning to different therapeutic agents to identify the most effective and well tolerated drug. In contrast, 86% of the 7 controlled RA patients were on some form of immunomodulatory therapy at the time of knee aspiration. No RA patients in either group were receiving JAK inhibitors or leflunomide. As noted in Table 1, the number of RA patients receiving immunomodulatory drugs exceeded the total number of both RA active (*n* = 11) and RA controlled (*n* = 6) patients since many were on combination therapy. One OA patient was receiving hydroxychloroquine as a trial therapeutic by her treating rheumatologist for inflammatory osteoarthritis at the time of knee aspiration.

Interestingly, none of the three normal subjects had any detectable WBCs in their SF samples including one with a repeat aspiration at a different time, suggesting that SF from healthy knees may not contain WBCs. Also of interest, the mean SF WBC cell count of 131 cells/mm^3^ from the 21 OA patients was actually higher than the mean SF WBC level of 85 cells/mm^3^ from the RA patients with controlled disease.

The log concentration levels of our 16 separate cytokines, chemokines, MMPs and other proteins are presented in Figure 2 for our normal subjects and patients with OA, RA controlled and RA active. The first six panels in Figure 2a–f display concentration levels of TNF-α, MMP-, MMP-2, MMP-1, IL-1b and IL-7, which were statistically different between patients with OA (*n* = 21) compared to active RA patients (*n* = 20). None of the other 10 cytokines, chemokines, MMPs and proteins were significantly different between OA patients and the RA patients. This latter finding may also be related to the large variance in levels from several individual RA and OA patients. As noted for levels of IL-6 (panel i), IL-1 ra, (panel j) and IL-2 (panel m), some individual active RA patients had very low levels of these cytokines which most likely prevented between-group differences from reaching statistical significance. Interestingly, the means of the pro-inflammatory cytokines IL-6 (panel i), and IL-2 (panel m) as well as IL1-ra (panel j) show a progressive increase in levels comparing controls to OA to controlled RA and active RA patients. Also of interest is that normal healthy knees from this small cohort seemed to have higher SF IL-4 levels (panel n) than patients with RA or OA, and comparable levels of IL-18 (panel l), IL-15 (panel p), IGFBP-3 (panel g), IGFBP-4 (panel k), MMP-3 (panel h) and G-CSF (panel o), suggesting that some of these proteins may be involved in normal cartilage homeostasis. 

Table 2 displays the mean values, standard deviation and range for each analyte by each group of subjects. These are arranged from highest to lowest concentration for most analytes so that the individual levels of each cytokine, growth factor or MMP can more easily be compared than the log concentration utilized in Figure 2.

We also analyzed if any of these 16 proteins correlated with the SF WBC counts in our 27 RA patients with both active and controlled disease. As noted in Figure 3, the MMP-8 levels positively correlated with WBC levels, *p* < 0.001 in the active RA group. None of the other 15 SF proteins significantly correlated with SF WBC levels. Since 63% of all RA patients (17 of 27) were on some form of immunomodulatory therapeutics including biologics, treatment could have confounded the lack of correlation between some of these SF proteins and WBC counts.

Figure 4 displays paired SF and serum levels of these 16 proteins in a subset of 11 OA subjects obtained simultaneously. Significant differences (*p* < 0.05) were observed between levels in the synovial fluid compared to serum for IL-4, IL- 6, IL-8, IL-15, IL-18, MMP-8 and MMP-9. The discordance in serum vs. SF levels was greatest for IL-6, IL-8, IL-4 and IL -15. Therefore, measuring these cytokines and proteins in the peripheral blood is unlikely to serve as an accurate biomarker for levels within the synovial space or accurately reflect the inflammatory process within the joints of OA patients.

## 4. Discussion

In comparing the results of this SF biomarker study to other published reports, it is important to note that all our RA and OA patients had symptomatic knee pain sufficiently severe enough to require an IA therapeutic or a diagnostic aspiration. Therefore, the severity of knee pain in our patients could potentially affect some cytokine levels and protein levels compared to sampling of knee SF from patients with less severe pain. In addition, none of our patients had acute knee pain due to trauma or had received an IA injection within 3 months of their aspirations. Since only 1 of our 51 subjects, one with OA, had a BMI > 40, our results might also not be comparable to OA patients with morbid obesity, especially since obesity may be an independent risk factor for osteoarthritis due to immune-mediated mechanisms [19].

Our SF cytokine levels may also differ from studies of more advanced OA patients such as those who had SF obtained pre-operatively for total joint arthroplasty or surgical intervention following traumatic injuries requiring cruciate ligament, or meniscal reconstruction [15,20,21]. In a study of 34 OA patients prior to knee arthroplasty for unicompartmental or bicompartment knee pain, Nees et al. reported a significant correlation between radiographic OA severity and levels of IL-4, IL-6, IL-8, IFN gamma, SCGF-β, VEGF, and CXCLL01, whereas knee pain also correlated with levels of IFN gamma, SCGF-β, and VEGF but also with IL-7, IL-10, IL-12, and IL-13 levels [22]. Moradi et al. have demonstrated elevated pro-inflammatory cytokine profiles (CXCL1, exotoxin, IFN gamma, IL-7, IL-8, IL-9 and IL-12) in SF knee samples, which could help differentiate bicompartmental knee OA from unicompartmental involvement [23]. Larson et al. also reported that SF IL-6 and TNF-α levels in 132 patients who underwent meniscectomy 15–22 years previously and had rising levels of these cytokines in a subset of 71 subjects repeated 4–10 years later correlated with radiographic OA progression [24]. While we did not observe statistically significant differences in IL-6 levels between our four groups of patients, as demonstrated in Figure 2 (panel i), that was most likely due to the large variance noted in IL-6 levels from several of our OA and RA patients who had very low levels. However, we did observe a progressive increase in SF IL-6 levels between normal subjects, OA patients and controlled RA patients compared to active RA patients. We also speculate that the very low levels TNF-α, IL-6, and IL-2, in some RA patients noted in Figure 2 may reflect a true bimodal distribution of these cytokines being produced among different RA phenotypes. This may help explain the lack of responsiveness of individual RA patients to biologics which target different inflammatory or cytokine pathways. We also demonstrated that SF IL-4 levels were similar in our normal subjects, OA or RA patients which suggests this cytokine might be an anabolic cytokine as reported by Katz et al. [4].

SF levels of cytokines from patients with traumatic or meniscus injury are also unlikely to be comparable to our patients with non-traumatic or milder forms of OA. SF cytokine levels obtained from contralateral knees during surgery may also involve a dilution effect from the prior instillation of saline if obtained during arthroscopy [25,26]. In addition, intra-operative SF obtained from contralateral knees may not be “normal” but rather represent a milder and less painful phenotype of OA compared to the symptomatic operated knee. Our SF cytokine levels from normal subjects may also differ from samples obtained from cadavers since those may reflect post mortem changes. In contrast to the study by Hyaldahl et al., all our normal subjects were recreational runners, but SF was not aspirated pre and post running [27].

The fact that none of our normal subjects had WBCs in their SF is important new information as most clinical labs report SF WBC counts as abnormal if values exceed 25 to 150 WBCs/mm^3^. Many of our OA patients had WBC > 50 cells/mm^3^; however, none had SF WBC counts > 300 cells/mm^3^. Therefore, the presence of elevated numbers of WBCs in the SF in OA may also contribute to further cartilage degradation especially if the inflammatory milieu increases as the disease progresses. Elevated WBC counts in OA have been reported by McCabe et al., since 21 of their 55 knee OA patients had WBC counts between 250 and 1000 cells/mm^3^ and these counts correlated with synovitis on MRI and IA glucocorticoid responsiveness [28]. Another relevant recent publication by Rolle et al. also reported a mean SF WBC count of 341 cells/mm^3^ from the knees of 67 OA patients, whereas SF volumes correlated better with the severity of radiographic joint damage than SF WBC counts [29].

Our results in Figure 4 are similar to most prior studies that have measured simultaneous SF and peripheral blood cytokine levels and have also demonstrated poor correlation or much lower levels in the peripheral blood of both OA and RA patients. Catterall et al. measured simultaneous SF and serum biomarkers following acute ACL injury and reported that of the 20 biomarkers measured in the SF, only 12 could be accurately measured in the serum and of those, only 4 (CTxI, NTx, osteocalcin and MMP-3) correlated well with levels in the SF [21]. In another study of 150 patients within 8 weeks of an acute knee injury, elevated SF levels of IL-6, monocyte chemotactic protein 1, MMP-3, TIMP-1 Activin A and TSG-6 were detected which fell over the next 3 months, whereas simultaneous peripheral blood serum or plasma cytokine levels had either undetectable levels or correlated poorly to those in the SF [20]. In a knee lavage study of 23 non-RA patients with symptomatic meniscal tears and 15 normal subjects without knee pain, elevated levels of multiple SF cytokines were observed (IFN gamma, IL-2, IL-4, IL-6, IL-10, IL-13, MCP-1 and MIP-1-β) from injured knees compared to asymptomatic contralateral knees and from normal knees [15].

For SF cytokine analysis, we separated our RA patients into two groups based upon inflammatory disease activity (SF WBC < or > 300 cells/mm^3^ rather than by serologic phenotype. Gomez-Puerta reported differences in SF cytokine levels between antibodies to cyclic citrullinated peptide antigen (ACPA) positive and ACPA negative patients [30]. Our controlled RA patients (those with WBC < 300 cells/mm^3^) most likely had advanced degenerative changes despite reasonable control of their synovitis with biologics, cDMARD or combination therapy since 86% were receiving these therapeutic agents at the time of arthrocentesis.

When SF cytokine levels were measured simultaneously with peripheral blood levels from RA patients, again significant discordance was reported [31,32]. In a report involving 318 RA patients, a panel of 12 peripheral blood biomarkers including IL-6, TNF-α, MMP-1 and MMP-3 levels failed to correlate with standard disease activity scales (DAS 28-CRP, RAPID-3, CDAI) or predict responsiveness to abatacept or adalimumab [33]. In another study of 42 RA patients, 12 cytokines were measured in the SF using a multiplex assay and simultaneously measured plasma levels were lower than those in the SF [31]. While changes in some SF cytokines tended to predict responsiveness to TNF-α inhibitor therapy, plasma level changes were not observed after TNF inhibitor therapy. 

The ideal SF biomarker study to predict drug responsiveness in RA patients would be to obtain SF for cytokine analysis prospectively in patients with early RA prior to the beginning immunomodulatory drug therapy. A similar strategy using peripheral blood is underway in the UK, Maximizing Therapeutic Utility in RA, (MATURA) [34]. However, recruitment for this study design in the US would be challenging since most RA patients are treated early with MTX at the onset of their inflammatory arthritis or hydroxychloroquine if they have less aggressive disease. Nevertheless, identifying the SF cytokine profile for different phenotypes of RA patients has the potential to enhance precision medicine-based therapeutic strategies for an individual patient once a test has been validated. In an excellent recent review by Schett et al., they have outlined various cytokine pathways which are characteristic of the more common IMIDs based upon clinical trial experience with responsiveness or failure to respond to specific cytokine inhibitors in RA patients compared to patients with other IMIDs including psoriatic arthritis, juvenile idiopathic arthritis, axial spondyloarthritis, ulcerative colitis and Crohn’s disease [16].

This study does have several limitations in addition to small sample sizes, especially in the normal group. While we did have IRB and funding agency approval to study 10 healthy subjects, funding was halted when a required milestone to have approval to study US astronauts was not granted after a NASA Program office informed us that cartilage health was not listed as a Human Research Program-designated accepted risk to human space flight at that time. A potential confounding variable in our RA patients was the potential effects of drug therapy since 63% were on some form of immunomodulatory therapy. Immunomodulatory drug therapy most likely affected levels of some SF cytokines and various proteins among our RA patients and possibly the one OA patient receiving hydroxychloroquine. These therapeutic agents may also explain why none of the pro-inflammatory cytokines (TNF-α, IL-1-β, IL-6, IL-18 or IL-15) correlated statistically with WBC levels in our RA patients. Some of our variance in SF cytokine levels may also have been the result of a batch testing effect since these cytokines and MMP levels were measured at three different times in batches of 20 samples each except for the paired SF and serum samples which were run as a single batch on the same day. Variance in cytokine levels has been observed in other clinical labs when run at different times [35]. However, a strength of this study, is the uniformly short interval of 30 to 60 min between sample aspiration to cryopreservation. Cytokine analysis in biological samples such as serum and plasma is affected by delayed separation from cells, prolonged retention at room temperature or repeated freeze–thaw cycles [36,37]. The cytokine results that we report here may differ from those in other studies due to variances in sample handling and storage protocols.

In this study, we have also overcome some limitations of prior SF biomarker research from OA patients with small volumes of SF by obtaining adequate SF volumes for multiplex analysis using an external pneumatic compression device along with US guidance for aspiration. This enabled us to obtain adequate SF volumes even from some patients who had very small SF volumes measuring only 1–2 mm of vertical depth on US [38]. Bhavsar and Rolle et al. have also demonstrated that compression with a viscoelastic knee brace increased the volume of SF aspirated compared with aspiration without compression using landmark-based arthrocentesis [29,39]. While we have presented results from 16 different chemokines, cytokines, MMPs and other select proteins in this paper, we have also been able to quantitate over 50 separate proteins using multiplex bead analysis technology from SF cryopreserved samples with volumes between 0.5 and 1.0 mL.

## 5. Conclusions

The results of this SF biomarker study revealed that only 6 of 16 proteins (TNF-α, IL-1-β, IL-7, MMP-1, MMP-2 and MMP-3) were significantly higher in active RA patients than OA patients, whereas levels of IL-1 ra, IL-2, IL-4, IL-6, IL-15, IL-18, MMP-3, G-CSF, IGFBP-3 and ILGFBP-4 did not different statistically. These findings suggest that OA patients with non-traumatic knee pain have a pro-inflammatory SF environment similar to many RA patients which may contribute to ongoing cartilage loss and ECM degradation. We also noted that only SF MMP-8 levels correlated with SF WBC levels in RA patients with active disease (SF WBC counts > 300 cells/mm^3^), whereas the other 15 proteins did not correlate statistically. We also observed that there was significance discordance between 7 of these 16 biomarker protein levels in the SF (IL-4, IL-6, IL-8, IL-15, IL-18, MMP-8 and MMP-9) when compared to levels in the peripheral blood obtained simultaneously in a subset of 11 OA patients. Therefore, measurement of cytokines and other key biomarkers in SF is probably more likely to yield a validated panel of wet biomarkers to assess cartilage health and responsiveness to drug therapy than relying on circulating levels in the peripheral blood. Finally, we hope that using external compression with ultrasound guidance and multiplex testing will facilitate SF-based biomarker research and advance precision medicine by identifying the optimum disease-modifying therapeutic for a specific arthritis patient [40].

## Figures and Tables

**Figure 1 jcm-10-05027-f001:**
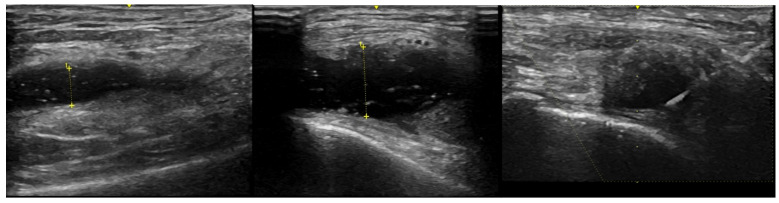
US images from a single OA patient displaying the maximum synovial fluid (anechoic) region from the lateral knee. The left panel was baseline before external pneumatic compression (SF depth was 5.8 mm), middle panel was after inflation to 100 mmHg (SF depth was 10.9 mm) and the right panel is of needle insertion during aspiration while the device remained inflated.

**Figure 2 jcm-10-05027-f002:**
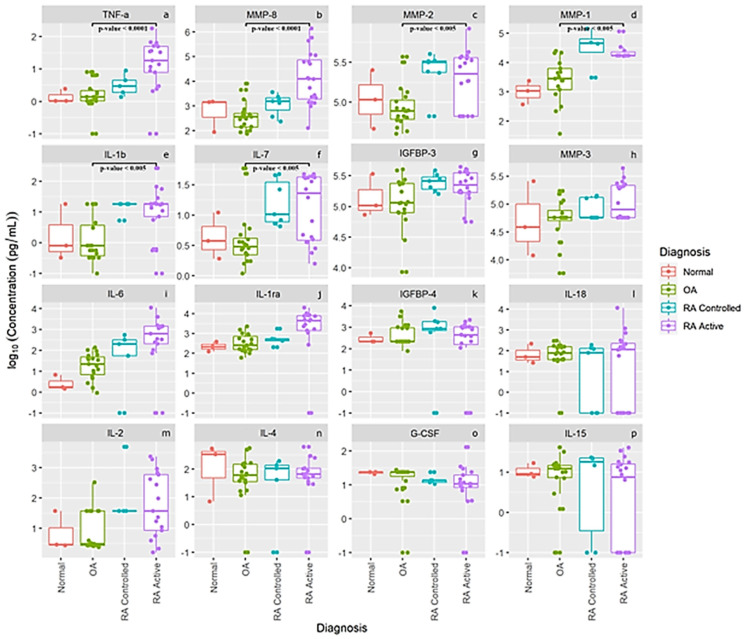
Synovial fluid log concentration levels of cytokines, chemokines, MMPs and select protein levels from normal subjects, patients with OA, and patients with controlled rheumatoid arthritis and active rheumatoid arthritis. Panel legend: panel (**a**) TNF-α = Tumor Necrosis Factor alpha, panel (**b**) MMP-8 = Matrix Metalloproteinase 8, panel (**c**) MMP-2 = Matrix Metalloproteinase 2, panel (**d**) MMP-1 = Matrix Metalloproteinase 1, panel (**e**) IL-1 b = Interleukin 1 beta, panel (**f**) IL-7 **=** Interleukin 7, panel (**g**) IGFBP-3 = Insulin-Like Growth Factor-Binding Protein 3, panel (**h**) MMP-3 **=** Matrix Metalloproteinase 3, panel (**i**) IL-6 = Interleukin 6, panel (**j**) L-1 ra = Interleukin 1 receptor antagonist, panel (**k**) IGFBP-4 = Insulin-Like Growth Factor-Binding Protein 4, panel (**l**) IL 18 = Interleukin 18, panel (**m**) IL-2 = Interleukin 2, panel (**n**) IL-4 **=** Interleukin 4, panel (**o**) G-CSF **=** Granulocyte Colony Stimulating Factor, and panel (**p**) IL-15 **=** Interleukin 15.

**Figure 3 jcm-10-05027-f003:**
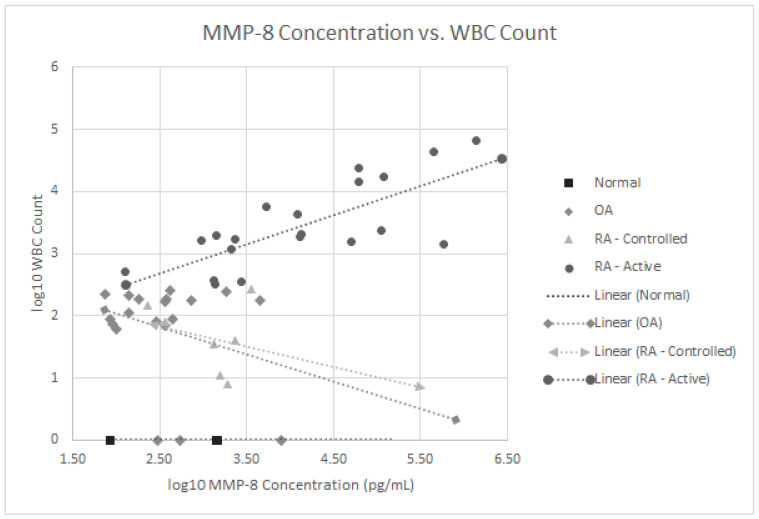
Log 10 plot of WBC counts vs. MMP-8 levels in normal subjects, patients with osteoarthritis, controlled rheumatoid arthritis and active rheumatoid arthritis.

**Figure 4 jcm-10-05027-f004:**
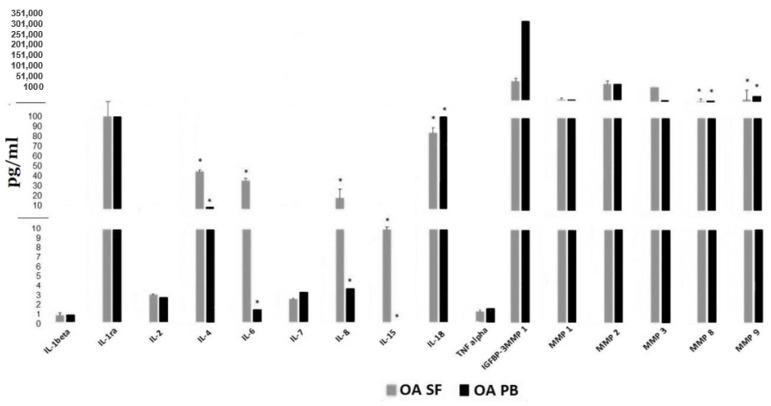
Cytokine and MMP levels in synovial fluid compared to simultaneous serum levels from 11 OA patients. * indicates statistical differences, *p* value < 0.05 between SF (synovial fluid) cytokine or MMP levels and those in the PB (peripheral blood).

**Table 1 jcm-10-05027-t001:** Demographics and immunomodulatory drug use among patients with RA and OA as well as normal subjects.

	RA Active	RA Controlled	OA	Normal
Number of Subjects	20	7	21	3
Age Range (years)	33–75	49–78	39–88	44–68
Mean Age	55	63	63	57
Gender (F vs. M)	15(75%)/5(25%)	7(100%)/0(0%)	11(52%)/10(48%)	2(66%)/1(33%)
BMI				
Range	20–35	17–37	23–41	21–24
Mean	26	27	30	22
Number on Prednisone, DMARD or Biologic	11 (58%)	6 (86%)	1 (5%)	0
Prednisone	6 (32%)	1 (14%)	0	0
Infliximab	3 (16%)	0	0	0
MTX	2 (11%)	2 (29%)	0	0
HCQ	3 (16%)	3 (43%)	1	0
Rituximab	2 (11%)	1 (14%)	0	0
Tocilizumab	1 (5%)	1 (14%)	0	0
Sulfasalazine	1 (5%)	0	0	0
Etanercept	1 (5%)	0	0	0
Adalimumab	1 (5%)	0	0	0
Golimumab	0	1 (14%)	0	0
+RF > 14	7 (35%)	6 (86%)	ND	ND
+CCP > 20	10 (50%)	6 (86%)	ND	ND
SF WBC (cells/mm^3^)				
Range	331–65,000	8–270	0–260	0
Mean	9620	85	131	0

**Legend:** RA = rheumatoid arthritis, OA = osteoarthritis, Normal = asymptomatic recreational runners without knee pain, RA active = patients with SF WBCs > 300 mm^3^, RA controlled = patients with SF WBCs < 300 cells/mm^3^, BMI = body mass index, MTX = methotrexate, HCQ = hydroxychloroquine, +RF = rheumatoid factor with an antibody titer > 14, and +anti-CCP = antibodies to cyclic citrullinated peptide antigens with a value of > 20 units.

**Table 2 jcm-10-05027-t002:** Displays the mean values, standard deviation and range for each analyte in pg/mL for normal subjects, patients with osteoarthritis, compared to patients with controlled rheumatoid arthritis and active rheumatoid arthritis.

	Normal		OA		RA Controlled		RA Active	
Protein	Mean +/− SD	Range	Mean +/− SD	Range	Mean +/− SD	Range	Mean +/− SD	Range
IGFBP-3	172,185 +/− 146,424	72,675–340,318	161,179 +/− 113,010	8501–400,152	256,352 +/− 82,938	158,808–388,506	247,003 +/− 115,594	55,900–441,896
MMP-2	135,886 +/− 106,161	46,625–253,279	104,914 +/− 86,836	40,305–370,590	273,587 +/− 107,471	66,885–400,806	251,110 +/− 191,950	66,885–833,826
MMP-3	102,994 +/− 135,473	11,821–258,664	67,820 +/− 46,784	5671–172,394	89,966 +/− 41,527	56,800–140,274	147,129 +/− 110,500	56,800–443,737
MMP-1	1259 +/− 994	370–2331	6300 +/− 8107	36–24,804	60,084 +/− 60,225	3041–145,083	29,601 +/− 30,809	16,569–115,877
MMP-8	999 +/− 790	86–1478	1000 +/− 1977	74–7978	1619 +/− 1162	232–3605	141,947 +/− 325,035	127–1,358,400
IGFBP-4	320 +/− 183	215–531	825 +/− 1292	78–5384	1983 +/− 2727	0–7988	657 +/− 618	0–2167
IL-4	294 +/− 268	6–536	119 +/− 154	0–563	95 +/− 77	0–194	129 +/− 156	0–618
IL-1ra	241 +/− 131	122–381	527 +/− 581	60–2254	681 +/− 599	200–1727	5021 +/− 5032	0–19800
IL-18	97 +/− 104	25–216	103 +/− 87	0–302	75 +/− 76	0–186	745 +/− 2545	0–11,486
G-CSF	23 +/− 2	20–24	20 +/− 8	0–27	15 +/− 5	10–24	20 +/− 29	0–128
IL-2	14 +/− 20	2–37	29 +/− 74	2–327	988 +/− 2128	37–4794	429 +/− 694	1–2320
IL-15	11 +/− 5	7–16	12 +/− 10	0–40	14 +/− 11	0–22	10 +/− 12	0–40
IL-7	6 +/− 5	1–11	8 +/− 16	1–59	22 +/− 18	6–47	23 +/− 19	1–47
IL-1b	6 +/− 10	0–18	5 +/− 7	0–18	15 +/− 6	5–18	33 +/− 62	0–263
IL-6	3 +/− 3	1–6	36 +/− 39	0–135	225 +/− 221	0–550	1662 +/− 2763	0–11190
TNF-α	1 +/− 1	1–2	2 +/− 3	0–8	4 +/− 3	1–8	34 +/− 43	0–176

## Data Availability

The data generated in Figure 2, Figure 3 and Figure 4 were entered into the REDCap database at National Jewish Health and contain PHI so they are considered confidential except for access by the investigators and biostatistician James Crooks who performed the statistical analysis.

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
