# Peer review of "Synovial Fluid Cytokines, Chemokines and MMP Levels in Osteoarthritis Patients with Knee Pain Display a Profile Similar to Many Rheumatoid Arthritis Patients"

_jcm, 2021, doi:10.3390/jcm10215027_

Round 1
Reviewer 1 Report
Modify the abstract: background not provided. Aim of the study have to be better explained.
Also int the introduction section the aim of the study have to be provided, because it is very important to explain why the study has been conducted and what Authors want to demonstrate.
Methods section seems too long. Try to reduce the lenght of the text, providing only necessary information.
In the conclusion I suggest to report only the main findings of the study, to have a shorter paragraph
Author Response
Reviewer number 1
- Regarding adding the background to the abstract section has been completed (line 16-20).
- Regarding adding study aims to the introduction section, as per the other reviewer that has been added as the last paragraph (line 81-87).
- The methods section has now been revised eliminating or reducing text in 8 sentences. We wanted to keep the sample handling and assay methods as is, so that other investigators can use identical procedures to avoid large variations in cytokine level measurements as we detailed in the discussion section.
- Regarding the conclusion section, we have re-written this as bullet statements to highlight main findings of the study.
Reviewer 2 Report
I personally congratulate the authors for their work. It is really a necessity to better understand and obtain effective treatments for OA.
I only wish they could modify and clarify the following points:
- In the title, correct Dis-Play.
- Although it seems to be clear at the end of your introduction, I recommend the authors to write the last sentence of this section as a clear research objective.
- Clarify why the asymptomatic group was so small (N=3).
- Figure 2 and 4 are difficult to visualize and interpret.
- I recommend you to rewrite Disscusion, synthesizing in bullet points or in no more than two sentences, the main findings of the study.
Author Response
We greatly appreciate the kind congratulations statement of this reviewer.
Regarding suggested modifications
- Title has now been corrected, some of those errors might have been a format error.
- Regarding modifying the introduction section, we agree with this very helpful suggestion and the aims of the study have now been added as the last 3 sentences of the introduction section line 81 to 87 of the revised manuscript.
- We also regret we did not have a larger number of normal subjects as the original CASIS/NASA study was approved for 10 but the funding stopped after one of the milestones was not achieved when the International Space Station Program Science Office did not approve our using Astronauts as study subjects since the NASA HRP (Human Research Program) Office did not list Cartilage Health as a “current accepted risk to human space flight”. Subsequently, 6 ISS astronauts have needed Total joint arthroplasty so “Cartilage Health in microgravity” is now being added to Bone health so that this type of research can be approved. Hopefully our original CASIS (Center for Advancement of Science in Space) grant can be approved for funding to resume this research on US Astronauts as well as normal control subjects. I have added a brief “diplomatic” explanation in the discussion section (line348-352) so as not to offend the ISS Program Science Office.
- regarding Figure 2 and 4 I have added some additional clarification in the discussion section to aide with interpretation.
5. Regarding the Discussion Section. I assume this reviewer meant the Conclusion section rather than the discussion section so the Conclusion section was been modified with bullet points as suggested.